# In the Dawn of an Early Invasion: No Genetic Diversity of *Angiostrongylus cantonensis* in Ecuador?

**DOI:** 10.3390/pathogens12070878

**Published:** 2023-06-27

**Authors:** Luis Solórzano Álava, Cesar Bedoya Pilozo, Hilda Hernandez Alvarez, Lazara Rojas Rivera, Misladys Rodriguez Ortega, Jorge Fraga Nodarse, Leandro de Mattos Pereira, Raquel de Oliveira Simões, Roberto do Val Vilela

**Affiliations:** 1Hospital Luis Vernaza, Junta de Beneficencia de Guayaquil, Guayaquil 090101, Ecuador; lsolorzano@jbgye.org.ec (L.S.Á.); cbedoya@jbgye.org.ec (C.B.P.); 2Instituto de Medicina Tropical Pedro Kouri, La Habana 17100, Cuba; hilda@ipk.sld.cu (H.H.A.); lrojas@ipk.sld.cu (L.R.R.); misladys@ipk.sld.cu (M.R.O.); fraga@ipk.sld.cu (J.F.N.); 3Instituto Tecnológico Vale, Belém 66055-090, PA, Brazil; leandro.mattos@pq.itv.org; 4Departamento de Parasitologia Animal, Instituto de Veterinária, Universidade Federal Rural do Rio de Janeiro, Seropédica 23890-000, RJ, Brazil; raquelsimoes@ufrrj.br; 5Instituto Oswaldo Cruz, Fundação Oswaldo Cruz, Rio de Janeiro 21040-900, RJ, Brazil

**Keywords:** Invasive species, eosinophilic meningoencephalitis, cytochrome c oxidase subunit I

## Abstract

The nematode *Angiostrongylus cantonensis* has been reported worldwide. However, some basic questions remain unanswered about *A*. *cantonensis* in Ecuador: (1) Was the invasion of *A*. *cantonensis* in Ecuador unique, or did it occur in different waves? (2) Was this invasion as recent as historical records suggest? (3) Did this invasion come from other regions of South America or elsewhere? To address these issues, we assessed the genetic diversity of MT-CO1 gene sequences from isolates obtained in 11 of Ecuador’s 24 provinces. Our Bayesian inference phylogenetic tree recovered *A*. *cantonensis* as a well-supported monophyletic group. All 11 sequences from Ecuador were identical and identified as AC17a. The haplotype AC17a, found in Ecuador and the USA, formed a cluster with AC17b (USA), AC13 (Thailand), and AC12a-b (Cambodia). Notably, all the samples obtained in Ecuadorian provinces’ different geographic and climatic regions had no genetic difference. Despite the lack of genetic information on *A*. *cantonensis* in Latin America, except in Brazil, our finding differs from previous studies by its absence of gene diversity in Ecuador. We concluded that the invasion of *A*. *cantonensis* in Ecuador may have occurred: (1) as a one-time event, (2) recently, and (3) from Asia via the USA. Further research should include samples from countries neighboring Ecuador to delve deeper into this.

## 1. Introduction

The rat lungworm *Angiostrongylus cantonensis* (Chen, 1935) was first described in the bronchi of the rodents *Rattus rattus* (Linnaeus, 1758) and *Rattus norvegicus* (Berkenhout, 1769) in Guangzhou (formerly Canton), China [1]. This nematode is the etiological agent of neuroangiostrongyliasis, which is the leading cause of eosinophilic meningitis (EM) or eosinophilic meningoencephalitis (EME) in humans, an infectious disease of the central nervous system [2]. This disease is characterized primarily by increased eosinophils in peripheral blood and cerebrospinal fluid, among other symptoms such as fever and severe headache [3,4].

The first documented human case of neuroangiostrongyliasis was in Taiwan in 1944. However, it took nearly two decades to establish a clear link between the parasite and the disease (i.e., *A*. *cantonensis* as a causative agent of EME) [5]. Since the first report, several outbreaks have been reported globally as the parasite has spread from traditional endemic regions of Southern China and Southeast Asia to the Pacific islands, Japan, Australia, Africa, the Canary Islands, the Balearic Islands, and the Americas, including the USA, Caribbean islands, and Brazil [2,4,6,7]. In Europe, *A*. *cantonensis* infections have been reported in different countries, although apparently, only one case was autochthonous [8]. By 2008, more than 2800 cases of human angiostrongyliasis had already been recorded in 30 countries [9]. The spread of parasites in different regions threatens people living in endemic areas and a growing number of travelers visiting these regions [10].

In 2008, *A*. *cantonensis* was reported for the first time in Ecuador parasitizing the giant African land snail *Achatina* (*Lissachatina*) *fulica* Bowdich, 1822 and the rats *R*. *rattus* and *R*. *norvegicus* [11,12]. Since then, outbreaks and isolated cases have been reported to the Ministry of Public Health of Ecuador (MSP) [13], with most clinical-epidemiological suspicion and one necropsy-confirmed case [14]. The parasite is now considered endemic throughout most of the country [11,15]. The invasive pest *A*. *fulica* is one of the main intermediate hosts for *A*. *cantonensis* [16]. This mollusk lives in urban and rural areas and plays a vital role in the spread of the parasite [17]. Humans may become infected by ingesting food contaminated with third-stage larvae or eating infected raw snails [2]. Thus, *A*. *fulica* is an essential transmitter of eosinophilic meningoencephalitis and ocular angiostrongyliasis [18].

Different molecular biology methods have been employed to detect *A*. *cantonensis* [19,20,21,22,23,24]. Furthermore, they have been applied to explore systematic and population genetic aspects of *Angiostrongylus* taxa since populations have significant variability [25,26,27,28,29,30,31,32]. The use of mitochondrial genes, such as cytochrome c oxidase subunit I (MT-CO1), as molecular markers for specific parasite identification has been efficient [33,34,35,36]. The MT-CO1 gene has been used in studies on phylogeny, phylogeography, and haplotype identification [37,38,39,40]. However, some basic questions remain unanswered about *A*. *cantonensis* in Ecuador: (1) Was the invasion of *A*. *cantonensis* in Ecuador a single event, or did it occur in different waves? (2) Was this invasion as recent as historical records suggest? (3) Did this invasion come from other regions of South America or elsewhere?

To tackle these questions, we assessed the genetic diversity of MT-CO1 gene sequences from isolates obtained in 11 of Ecuador’s 24 provinces. Thus, we verified how many lineages could be found in different regions of Ecuador and whether there was enough time for the lineages to diversify. We also established these isolates’ phylogenetic and phylogeographic relationships, comparing them with other sequences from South America and the rest of the world. Consequently, we could retrace the possible origin of the lineages found in Ecuador.

## 2. Materials and Methods

### 2.1. Parasites and Experimental Infection

Third-stage larvae (L_3_) were obtained from *A*. *fulica*, collected in 11 provinces of Ecuador (Figure 1) using the catch-per-unit-effort method for 30 min in each locality [16]. The L_3_ were used to experimentally infect 12-week-old adult female Wistar strain *R*. *norvegicus* rats (200 ± 2 g body mass). The Instituto de Investigación en Salud Pública (INSPI) vivarium supplied the rats with their corresponding health and genetic quality certificates. The cycle was maintained in the National Reference Center for Parasitology. An average of 150 L_3_ (counted in a Neubauer chamber) were orally administered to each rat using a pipette. Infected rats were separated into cages (two specimens per cage) and identified according to the locality (province) where the infected gastropods were collected. Rats were kept under controlled temperature (21–24 °C) and humidity (60%), alternating 12-h light/dark cycle, and received food and water at pH 7.0. All procedures followed the guidelines for the maintenance and use of laboratory animals, following the specific legislation covering animals used for scientific purposes Directive 2010/63/EU as amended by European Union (EU) Regulation 2019/1010 [41].

Thirty-five days after administration of the larvae, the rats were euthanized using CO_2_. The thoracic cavity (heart, pulmonary arteries, and lungs) was examined for parasitic nematodes (juvenile or adult) according to protocols previously established at INSPI [11]. Specific taxonomic characteristics such as caudal bursa and the spicule length were used to identify the nematodes [42,43]. Approximately 30–40 adult *A*. *cantonensis* specimens from two infected rats representing each province were stored in a sterile labeled 50 mL Falcon tube with 70% ethanol in an ultra freezer at −80 °C.

### 2.2. Molecular Phylogenetic and Phylogeographic Analyses

We used DNA sequences obtained from adult parasites, as previously reported [39,44,45], to conduct phylogenetic and phylogeographic studies. Genomic DNA samples were isolated from adult parasites recovered from the rats representing each province. Before DNA isolation, the nematodes were partitioned into tiny pieces with a scalpel and suspended in saline (0.9% NaCl). We used the QIAamp DNA Mini Kit (QUIAGEN, Hilden, Germany) for DNA isolation according to the manufacturer’s protocol. Each isolated DNA sample was identified according to its origin and stored at −80 °C until further amplification by PCR technique. Genomic DNA concentration was measured directly in a NanoDrop 2000 spectrophotometer (Thermo Fisher Scientific, Walthan, MA, USA).

DNA isolated from approximately 30 adult parasites was subjected to PCR to amplify the mitochondrially encoded cytochrome c oxidase I (MT-CO1) gene [37]. PCR reactions were performed in a 25 μL total volume containing 12.5 μL of GoTaq Colorless Master Mix (Promega, Madison, WI, USA: 2× DNA polymerase, 400 µM dATP, 400 µM dGTP, 400 µM dCTP, 400 µM dTTP, and 3 mM MgCl2, pH 8.5); 1.5 μL of 10 μM each MT-CO1 gene primer (Thermo Fisher Scientific, Walthan, MA, USA); 5.5 μL of distilled water; and 4 μL of genomic DNA. We also used a positive control consisting of an adult parasite DNA obtained from a wild-type rat (*R*. *rattus*) and a negative control with ultrapure water. The primers used were:

co1-F (5′TAAAGAAAGAAAGAACATAATGAAAATG3′)

co1-R (3′TTTTTTTTTTGGCATTCCTGAGGAGGT5′)

Modifications have been made to the original thermal cycling protocol by Vitta et al. [37] to standardize the technique in the INSPI laboratory and obtain the desired amplicons of approximately 450 base pairs (bp) as follows: 94 °C for 5 min; followed by 30 cycles of 94 °C for 1 min, 48 °C for 30 s, and 72 °C for 60 s; with a final extension at 72 °C for 5 min. PCR was performed in a C1000 Touch thermal cycler (Bio-Rad, Hercules, CA, USA).

We verified PCR products after 1.2% agarose (Promega, Madison, WI, USA) gel electrophoresis in 0.04 M Tris-acetate running buffer, 1 Mm ethylenediamine tetraacetic acid, pH 8.0 (Thermo Fisher Scientific, Walthan, MA, USA). We added 10 µL of Syber^®^ 1× (10,000×) dye (Thermo Fisher Scientific, Walthan, MA, USA) to the agarose gel. We added Blue/Orange Loading Dye, 6× (Promega, Madison, WI, USA), to each sample. TrackIt 100 bp DNA Ladder (0.1 µg/µL), with 100 to 1000 bp range (Thermo Fisher Scientific, Walthan, MA, USA), was used as molecular weight marker. Electrophoresis was performed at 80 V for 55 min using a PowerPac HC power supply (Bio-Rad, Hercules, CA, USA). PCR products were visualized using the ChemiDoc XRS imaging system (Bio-Rad, Hercules, CA, USA).

Amplicons purification; cycle-sequencing of both strands via the Sanger method, using the abovementioned PCR primers; and product precipitation, formamide resuspension, and analysis using the 3130 DNA Analyzer (Applied Biosystems, Foster City, CA, USA) were performed at the biochemistry department of the Universidad de las Américas (Ecuador).

The resulting chromatograms were edited with the software platform Geneious R7.0 (Biomatters, Aukland, New Zealand) [44]. Sense and anti-sense sequences of each amplified and sequenced sample were assembled into contigs. The resulting consensus sequences corresponding to 11 Ecuadorian provinces were deposited in the GenBank (Table 1).

To construct our MT-CO1 dataset, we used *A*. *cantonensis* sequences found in GenBank that overlapped ours (Appendix A). As outgroups, we added one sequence of *Angiostrongylus mackerrasae* Bhaibulaya, 1968 (MN793157) and three sequences of *Angiostrongylus malaysiensis* Bhaibulaya and Cross, 1971 (KT947979, KU532150, KU532153), all from GenBank (Appendix A). Sequences in the dataset were aligned by multiple alignments using MUSCLE [45], under default parameters, within the Geneious package. Final manual trimming of non-overlapping regions of the alignment was carried out using the Mesquite 3.70 software package [46].

Two different matrices were used in this study. In the first matrix, used for phylogenetic inferences, we excluded all duplicated sequences, keeping only one copy of each haplotype of *A*. *cantonensis* and the outgroup. In the second matrix, used for phylogeographic analyses, we included all *A*. *cantonensis* sequences and excluded the outgroup. To find the optimal partition clustering arrangements and corresponding log(ml) values in both matrices, we conducted Bayesian clustering of linked molecular data analyses using BAPS 6.0 [47,48].

Bayesian inference (BI) phylogenetic analyses were conducted using MrBayes 3.2.6 [49] on XSEDE within the CIPRES Science Gateway [50]. We used independent GTR+I+G models for each codon position, with unlinking of base frequencies and parameters. Sampling was performed by MCMC, for 10,000,000 generations, with four simultaneous chains, in two runs. Node supports were given by Bayesian posterior probabilities (BPP) of trees sampled every 100 generations after removal of the first 25% ‘burn-in’ generations. We assessed sampling adequacy using the program Tracer 1.7.1 [51] to calculate the effective sample sizes (ESSs) of parameters. We considered robust values above 200 effectively independent samples.

An intraspecific phylogeographic network was inferred using the program PopART, version 1.7 [52], with the median-joining network method [53]. Using DnaSP 6.12.03 [54], we organized the sequences into groups according to their geographic localities (countries). We also calculated, using DnaSP, the genetic diversity by the numbers of haplotypes (H), polymorphic sites (S), haplotype diversity (Hd), and nucleotide diversity (π). We finally used DnaSP for neutrality tests Tajima’s D [55] and Fu’s Fs [56].

## 3. Results

Along with our 11 MT-CO1 gene sequences of *A*. *cantonensis* from Ecuador, we added 105 sequences of *A*. *cantonensis* from GenBank and four sequences of outgroups. The full dataset had 120 sequences of *Angiostrongylus* ranging from 255 to 1617 bp in length (Appendix A). The haplotypes were named AC1-17, following the names for haplotypes previously adopted [38,40,57], adding letters to variants. All 11 sequences from Ecuador were identical and identified as AC17a. The Ecuador sequences were identical to five sequences from New Orleans, Louisiana, USA (USA-LA), retrieved from GenBank.

### 3.1. Molecular Phylogenetic Analyses

After multiple sequence alignment, trimming, and removal of all duplicates in the first matrix for phylogenetic inferences, the matrix resulted in 29 taxa and 255 sites. Of these, 201 were constant characters, and 41 were variable parsimony-informative characters. *Angiostrongylus cantonensis* was represented by 25 sequences, while the outgroup by four. According to the population structure recovered using BAPS, *Angiostrongylus* specimens were distributed in five clusters in the 29 sequences matrix.

After 25% burn-in removal, the BI mean estimated marginal likelihood was −751.4969, and the median was −751.1709. The ESS values were well above 200 for all parameters. The BI phylogenetic tree (Figure 2) recovered *A*. *cantonensis* as a well-supported monophyletic group (BPP = 1.00). Within *A*. *cantonensis*, the sequence AC17a, from Ecuador and USA-LA, was in a polytomy with AC17b (USA-LA); AC5a (Brazil, Japan, French Polynesia, and Hawaii, USA); AC5b (Japan); AC13 (Thailand); and a moderately supported clade (BPP = 0.70), formed by sequences AC8a (Brazil) and AC8b (Australia, Balearics, Canaries, Taiwan, and USA-LA). This polytomy was moderately supported (BPP = 0.78) and formed another polytomy with sequences AC12a and AC12b from Cambodia. This more inclusive polytomy was strongly supported (BPP = 0.98) and coincided with Cluster 3.

### 3.2. Phylogeographic Analyses

The second matrix, for phylogeographic analyses, included only sequences of *A*. *cantonensis*. This dataset included 11 sequences from Ecuador and 105 sequences from GenBank, excluding the outgroup, totaling 116 taxa and 255 sites after multiple sequence alignment and trimming. The total number of sites, excluding sites with gaps or missing data, was 254. The number of haplotypes was H = 25, the number of polymorphic sites S = 36, the haplotype diversity Hd = 0.895, the nucleotide diversity π = 0.02546, Fu’s Fs = −2.380, and Tajima’s D = −0.42728 (*p* > 0.10).

According to the population structure recovered using BAPS, *A*. *cantonensis* specimens were distributed in seven clusters in the 116 sequences matrix. We indicated the clusters in the intraspecific phylogeographic network (Figure 3). The haplotype AC17a, from Ecuador and USA-LA, formed a cluster with AC17b (USA-LA), AC13 (Thailand), and AC12a-b (Cambodia). This haplogroup was labeled Cluster 5 in the network.

## 4. Discussion

Introducing non-native mollusks, such as *A*. *fulica*, is essential in transmitting *A*. *cantonensis* [58]. Since the mid-20th century, *A*. *fulica* has been introduced into the tropics and subtropics and is considered the most harmful snail pest in these regions [17]. In Brazil, these mollusks were possibly introduced more than once on different occasions [59]. The first account is from the mid-1970s in the state of Minas Gerais [60]. The second, better documented, and probably the chief introduction was in the late 1980s in the state of Paraná from specimens brought from Indonesia for commercial purposes (snail farming) that were unsuccessful [61]. The giant African snail is now widespread in all 26 Brazilian states and the Federal District [62,63].

According to data from an Ecuadorian government organization, these snails were brought into the country in the mid-1990s. As in Brazil, this was for commercial purposes. Snail farms were built in some valleys of the Ecuadorian highlands, which offered an ideal temperature between 17 °C and 25 °C [64]. However, their breeding did not provide the expected economic returns. Inevitably, most of the farms were abandoned, and the snails were released into the environment. The result was a widespread infestation of urban and rural areas in almost all of Ecuador’s provinces [12]. *Achatina fulica* was probably the vector that introduced *A*. *cantonensis* to the country, as in Brazil [65] and China [66].

As for the definitive hosts, it is presumed that *R*. *rattus* arrived in Ecuador between the 16th and 17th centuries with the ships of the Spanish conquistadors [67]. *Rattus norvegicus* probably originated in China and spread to Europe, reaching North America through shipping during the second half of the 18th century. Both species are now widely distributed in urban areas worldwide [68].

The existence of intermediate and definitive hosts in almost all of Ecuador has contributed to the endemic nature of angiostrongyliasis distribution, making the control of this disease even more complex [11]. In 2008, the snail *A*. *fulica* (intermediate host) and the rat *R*. *rattus* (definitive host) were found naturally infected by *A*. *cantonensis* in Ecuador [11,12]. Both intermediate (*A*. *fulica*) and definitive (*R*. *rattus*) hosts are non-native species to Ecuador and are considered among the 100 most important invasive species in the world, according to the World Conservation Union [69]. Invasive species in an ecosystem can affect biotic alter interactions, impacting the economy, the environment, or public and animal health [70,71]. Moreover, elder Ecuadorians’ habit of eating raw snails increases the risk of *A*. *cantonensis* infection [12].

Earlier studies using the MT-CO1 to distinguish *A*. *cantonensis* isolates have shown different geographical isolates in determinate regions [38,39,40,57]. Tokiwa et al. [40] reported seven different haplotypes (AC1 to AC7): five were found in Japan (AC1, AC2, AC3, AC5, and AC7), two in mainland China (AC2 and AC6) and only one in Taiwan (AC1). In Brazil, analyses from 15 geographic isolates determined the presence of three different MT-CO1 haplotypes (AC5, AC8, and AC9). Most sample sequences were AC5 or AC8, whereas AC9 was a new haplotype [38]. Rodpai et al. [57] identified different *A*. *cantonensis* haplotypes in Cambodia, Myanmar, Thailand, and Hawaii, USA. Two of them (AC2 and AC5) had been previously reported. The AC2 haplotype, previously reported in China and Japan, was found in Myanmar. The AC5 haplotype, previously reported in Brazil and Japan, was found in Hawaii. Four new haplotypes (AC10-AC13) were also reported in Southeast Asia [57].

Such studies have shown that *A*. *cantonensis* in Asia has greater genetic diversity [39,40,57], indicating that this parasite has been circulating in these regions for a long time. Conversely, the sequence diversity of *A*. *cantonensis* is lower in many areas outside Asia [72]. Otherwise, there is little or no genetic information on the parasite in other regions of the planet, such as the Americas, except in Brazil [38].

In the present study, all sequences of the isolates from Ecuador were identical, the haplotype AC17a. In our phylogenetic analyses, this haplotype was nested into a polytomy with other sequences from around the world. Remarkably, all samples were obtained from provinces of Ecuador in different geographic and climatic regions, yet they did not show any genetic divergence between them.

The findings reported here represent a novelty in studying the genetic diversity of *A*. *cantonensis* isolates. Although there is a need for more information on the genetic diversity of this parasite in other Latin American countries, except for Brazil, our results are different from previous studies due to the complete absence of genetic diversity of *A*. *cantonensis* in Ecuador. Even admitting that the low number of nucleotide base pairs obtained could make the sequence homogeneous in the isolates from Ecuador, this same region of the MT-CO1 gene showed variations in the other haplotypes compared.

The fact that only one haplotype was found in 11 different Ecuadorian provinces is revealing. It strongly advocates a single introduction event. Furthermore, this result suggests that *A*. *cantonensis* has been recently introduced in the country, as there was no time for new haplotypes to differentiate from the original. This may justify the non-existence of genetic diversity among different circulating isolates.

Interestingly, the sequences from Ecuador shared a recent common ancestor with two Brazilian haplotypes (AC5 and AC8) [73]. However, it is unlikely that this could indicate a historical connection between the strains from both countries. The AC5 haplotypes found in Brazil from Pirituba (state of São Paulo), Queimados, and Niterói (state of Rio de Janeiro) correspond to a haplotype found in Japan, Hawaii, and French Polynesia [57,72,74], suggesting that the arrival of the parasite in Rio de Janeiro or São Paulo may have occurred from the Asian continent [38]. This hypothesis is also considered for the AC8a haplotype, closely related to AC8b, found in Australia, the Balearics, the Canaries, Taiwan, and the United States of America (USA). This shows the possible spread of *A*. *cantonensis*, with the giant African land snail, as a vector, from the arrival localities in Brazil to the Southeast, Northeast, and North Brazilian regions [38].

The sequences obtained here also grouped with AC13 and AC17 haplotypes from Thailand and the USA, respectively. The haplotypes AC10, AC11, and AC13, from Thailand, and AC12, from Cambodia, were described by Rodpai et al. [57] in phylogenetic studies using different DNA regions of *A*. *cantonensis* and *A*. *malaysiensis*. The haplotypes AC17, from the USA, were reported in a survey to identify *A*. *cantonensis* and determine the association between ecological characteristics and factors related to definitive hosts (*R*. *rattus*, *R*. *norvegicus*, *Sigmodon hispidus*, and *Oryzomys palustris*) associated with transmission risk of angiostrongyliasis in New Orleans [75]. The haplotypes AC12, AC13, and AC17 formed a cluster in the haplotype cluster analysis, suggesting that *A*. *cantonensis* may have arrived in Ecuador from Asia via the USA.

## 5. Conclusions

Our results suggest that the invasion of *A*. *cantonensis* in Ecuador occurred as a single event since only one haplotype was present in all 11 provinces studied, encompassing different ecoregions of Ecuador. Moreover, this invasion may have occurred recently, as we found no variation from the initial haplotype. It is unlikely that *A*. *cantonensis* reached Ecuador from Brazil. It is conceivable that the lineage found in Ecuador came from Asia via the USA. Future studies should sample countries neighboring Ecuador to infer migratory routes into this country in more detail.

## Figures and Tables

**Figure 1 pathogens-12-00878-f001:**
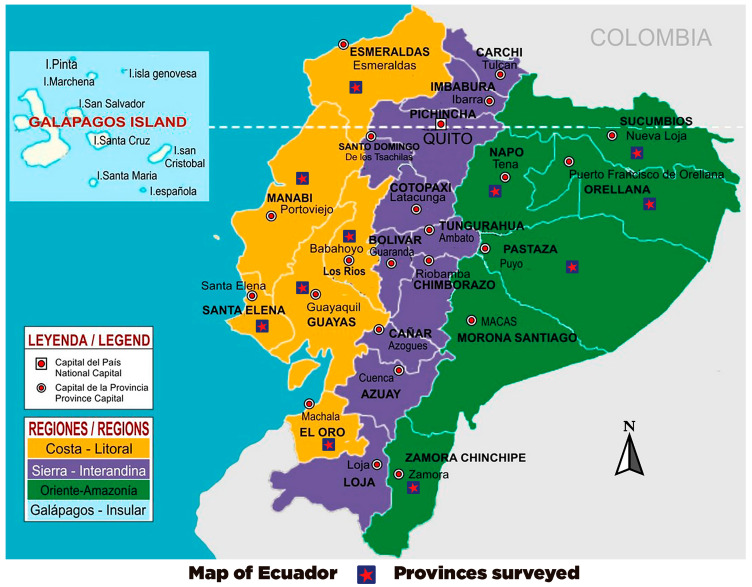
Map of Ecuador showing the study area highlighting the sampled provinces (https://provinciasecuador.com/mapa-politico-del-ecuador/ (accessed on 30 April 2023) with modifications).

**Figure 2 pathogens-12-00878-f002:**
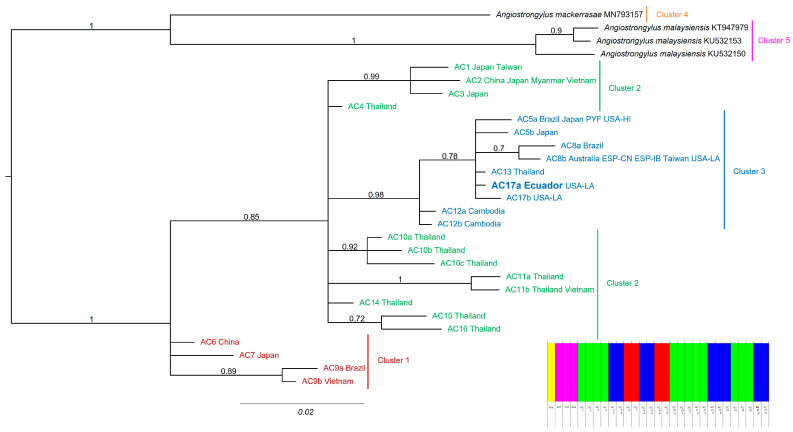
Bayesian inference (BI) phylogenetic relationships of *A*. *cantonensis* specimens and outgroups unique MT-CO1 gene sequences (255 bp). GenBank accession numbers of *A*. *cantonensis* sequences are provided in Appendix A. The values at the nodes are BPPs (>0.50). The scale bar is the number of substitutions per site. Sequence labels are colored based on the clusters recovered in the BAPS cluster analysis (bottom right). Sequences are labeled AC1-17, following names for haplotypes previously adopted [38,40,57], adding letters to variants, followed by the localities (countries) where they are found. Clusters 1–5 were recovered in the BAPS cluster analysis for the 29 sequences matrix.

**Figure 3 pathogens-12-00878-f003:**
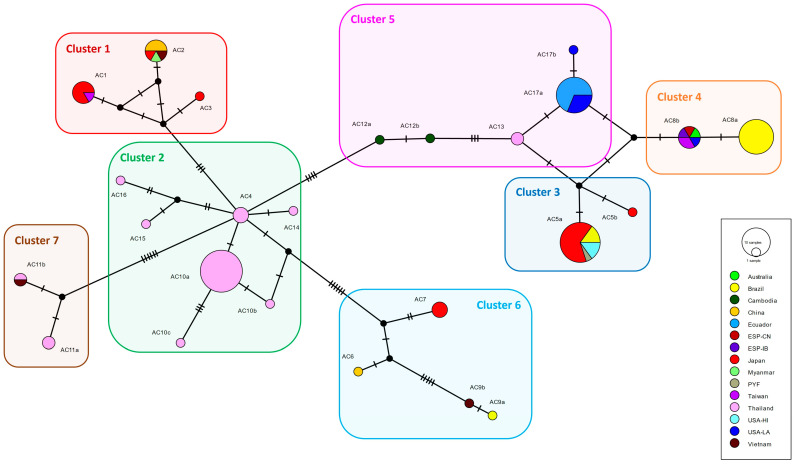
Median-joining haplotype network of *A*. *cantonensis* (25 haplotypes) based on 116 partial MT-CO1 gene sequences (255 bp). The size of the circles represents the frequency of haplotypes. The colors of the circles represent the localities (countries) of occurrence of each haplotype. Black circles are median vectors. Sequences are labeled AC1-17, following names for haplotypes previously adopted [38,40,57], adding letters to variants. Clusters 1–7 were recovered in the BAPS cluster analysis for the 116 sequences matrix.

**Table 1 pathogens-12-00878-t001:** Identification and GenBank accession numbers of sequences obtained in this study, followed by their respective sampling localities.

Identification	GenBank Accession Number	Province
LSA-01	MW391020	Esmeraldas
LSA-02	MW390970	Santa Elena
LSA-03	MW390971	El Oro
LSA-04	MW390972	Guayas
LSA-05	MW390967	Zamora
LSA-06	MW390974	Pastaza
LSA-07	MW390969	Orellana
LSA-08	MW390973	Manabi
LSA-09	MW390968	Napo
LSA-10	MW390966	Los Rios
LSA-11	MW390965	Sucumbios

## Data Availability

The data presented in this study are openly available in GenBank, accession numbers MW390965 to MW390974 and MW391020.

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
