# Peer review of "In the Dawn of an Early Invasion: No Genetic Diversity of *Angiostrongylus cantonensis* in Ecuador?"

_pathogens, 2023, doi:10.3390/pathogens12070878_

Round 1
Reviewer 1 Report
The paper entitled “In the dawn of an early invasion: No genetic diversity of Angiostrongylus cantonensis in Ecuador?” described a basic genetic information of the roundworm A. cantonensis in Ecuador. The author used only single gene, COI, and 11 samples for describing phylogeny. This is small number of samples. I understand that A. cantonensis samples are difficult to find. This manuscript is well written by concept between background of A. cantonensis in Ecuador and genetic diversity. I have no argument on manuscript that is well describe based on scientific logic and appropriate references. I have just a few comments.
1. Figure 2 BI phylogeny should be corrected.
All samples should demonstrate accession no. like in cluster 4 and cluster 5. Consider to use the bold letter instead of regular font for the sequences in the present study.
2. Page 7 in Table 1, please check “El Oro_” and “Pastaza_” underscore ?
English presented in the manuscript need minor correction.
Author Response
Dear Reviewer,
Thank you very much for your time spent and suggestions. We hope that the questions have been properly addressed by following your suggestions. We hope that the manuscript is clearer now and suitable for publication.
Please see below, in blue, for a point-by-point reply to comments and concerns.
Comments and Suggestions for Authors
The paper entitled “In the dawn of an early invasion: No genetic diversity of Angiostrongylus cantonensis in Ecuador?” described a basic genetic information of the roundworm A. cantonensis in Ecuador. The author used only single gene, COI, and 11 samples for describing phylogeny. This is small number of samples. I understand that A. cantonensis samples are difficult to find. This manuscript is well written by concept between background of A. cantonensis in Ecuador and genetic diversity. I have no argument on manuscript that is well describe based on scientific logic and appropriate references. I have just a few comments.
R: Thank you for your comments. Indeed, mitochondrial DNA, although a small fraction of the genomes of organisms, has been the most widely used marker in molecular diversity studies in metazoans for the last four decades, providing insight into inter and intraspecies diversity. Mitochondrial genes are markers of choice for population, phylogeographic, and phylogenetic studies because they are easy to amplify and manipulate, have high mutation rates, are putatively clonally inherited, and undergo mostly neutral evolution (Blouin, 2002; Galtier et al., 2009). The MT-CO1 gene represents a good marker for the study of the genetic evolution of Angiostrongylus cantonensis, in the differentiation of closely related Angiostrongylus species (phylogeny) and geographic isolates (phylogeography) of A. cantonensis, and in the identification of haplotypes (e.g., Monte et al., 2012; Vitta et al., 2016).
In 18 of Ecuador's 24 provinces, reports of Achatina fulica have been made. A. fulica, the intermediate host, is present in one of these provinces (Galapagos), but it is not infected with A. cantonensis (Llaguno et al., 2020). Six provinces from the coastal region (Manabi, El Oro, Los Rios, Esmeraldas, Santa Elena, and Guayas) and six from the Amazon region (Sucumbios, Napo, Orellana, Pastaza, Morona Santiago, and Zamora Chinchipe) were chosen as part of the effort to establish genetic differences. This is because the Andean Mountain Range naturally divides the two regions. Due to the demise of the definitive hosts in the province of Morona Santiago, adult A. cantonensis specimens could not be obtained for amplification and subsequent sequencing. Therefore, the study only included 11 provinces.
- Figure 2 BI phylogeny should be corrected.
All samples should demonstrate accession no. like in cluster 4 and cluster 5. Consider to use the bold letter instead of regular font for the sequences in the present study.
R: We appreciate this assessment. We have resized and bolded the font to highlight Ecuadorian sequences. Unlike clusters 4 and 5, each A. cantonensis sequence had from one to 22 accession numbers appended to it, as in the case of ac10a. Therefore, for simplicity, we have kept the accession numbers available in the supplementary material in Table S1 rather than overloading the figure with so many accession numbers.
We added to the Figure 2 subtitle the information: "GenBank accession numbers of A. cantonensis sequences are provided in the supplementary material Table S1".
Should we remove accession numbers from clusters 4 and 5 as well?
- Page 7 in Table 1, please check “El Oro_” and “Pastaza_” underscore ?
R: Thank you for noticing this. These were typos and have been deleted.
Comments on the Quality of English Language
English presented in the manuscript need minor correction.
R: Thank you for pointing this out. However, we are unsure of which minor correction is needed. We thoroughly revised our first draft using commercial proofreading programs. Nevertheless, we have now revised the manuscript again by using different editing tools. We hope the reviewer is satisfied and that the English language now complies with your high standards.
References
Blouin, M.S., 2002. Molecular prospecting for cryptic species of nematodes: mitochondrial DNA versus internal transcribed spacer. International Journal for Parasitology, ASP Special 32, 527–531. https://doi.org/10.1016/S0020-7519(01)00357-5
Galtier, N., Nabholz, B., Glémin, S., Hurst, G.D.D., 2009. Mitochondrial DNA as a marker of molecular diversity: a reappraisal. Molecular Ecology 18, 4541–4550. https://doi.org/10.1111/j.1365-294X.2009.04380.x
Llaguno, G., Álvarez, S.F., Moran, W., Colcha, A.C., Castro, R.R., Reinoso, G.L., Ormeño, C.Á., Soria, G.M., 2020. Salud Pública Veterinaria: Desde la docencia hacia la investigación. Grupo LUZUMA.
Monte, T.C.C., Simões, R. de O., Oliveira, A.P.M., Novaes, C.F., Thiengo, S.C., Silva, A.J., Estrela, P.C., Júnior, A.M., Cordeiro-Estrela, P., Maldonado-Júnior, A., 2012. Phylogenetic relationship of the Brazilian isolates of the rat lungworm Angiostrongylus cantonensis (Nematoda: Metastrongylidae) employing mitochondrial COI gene sequence data. Parasites & Vectors 5, 248. https://doi.org/10.1186/1756-3305-5-248
Vitta, A., Srisongcram, N., Thiproaj, J., Wongma, A., Polsut, W., Fukruksa, C., Yimthin, T., Mangkit, B., Thanwisai, A., Dekumyoy, P., 2016. Phylogeny of Angiostrongylus cantonensis in Thailand based on cytochrome c oxidase subunit I gene sequence. The Southeast Asian Journal of Tropical Medicine and Public Health 47, 377–386.

Reviewer 2 Report
The authors should update the bibliography in the introduction as, the parasite has been found in the Iberian Peninsula, Spain in addition to the Canary and Balearic Islands.
I suggest to the authors, for future studies, sequence more individuals from the same region paying attention to areas close to the coast.
I don't find the necessity to infect rats, why the authors don't sequence the L3 directly??
Author Response
Dear Reviewer,
We appreciate your time and advice very much. We trust that by taking your advice, the questions have been adequately answered. We believe the manuscript has improved and is now ready for publishing.
Please see below, in blue, for a point-by-point reply to comments and concerns.
Comments and Suggestions for Authors
The authors should update the bibliography in the introduction as, the parasite has been found in the Iberian Peninsula, Spain in addition to the Canary and Balearic Islands.
R: Thank you for bringing this up. The reviewer is right. We have added the following paragraph to the introduction:
"In Europe, A. cantonensis infections have been reported in different countries, although apparently, only one case was autochthonous (Federspiel et al., 2020).”
I suggest to the authors, for future studies, sequence more individuals from the same region paying attention to areas close to the coast.
R: Thank you for your comments, we intend to do that. The presence of Achatina fulica has been reported in 18 of the 24 provinces of Ecuador. One of these provinces (Galapagos) has the intermediate host A. fulica but it is not infected by A. cantonensis (Llaguno et al., 2020). Given the attempt to establish genetic differences, six provinces were chosen from the coastal region (Manabi, El Oro, Los Rios, Esmeraldas, Santa Elena, and Guayas) and six from the Amazon region (Sucumbios, Napo, Orellana, Pastaza, Morona Santiago, and Zamora Chinchipe), since both regions are separated by the natural geographical feature of the Andean Mountain Range. In the province of Morona Santiago, it was not possible to obtain adult specimens of A. cantonensis for amplification and subsequent sequencing because the definitive hosts died. So finally, only 11 provinces were included in the study.
I don't find the necessity to infect rats, why the authors don't sequence the L3 directly??
R: We appreciate this assessment. The adult parasite was used for two reasons: first, we wanted to ensure enough DNA, which was obtained from the adult parasite; second, we wanted to be sure that the larvae corresponded to A. cantonensis, and the way to confirm this was to observe the distinctive characteristics of the adult, whereas in L3 there is no certainty that they are A. cantonensis larvae or could belong to other species of the genus. The evidence that establishes the presence of the parasite is the finding of adult worms infecting the local rodent population (definitive hosts). Traditionally, this is done by searching for adult worms in the pulmonary arteries of rodents during necropsy and subsequent identification of species-specific morphological characteristics (Cowie, 2013).
References
Cowie, R.H., 2013. Biology, systematics, life cycle, and distribution of Angiostrongylus cantonensis, the cause of rat lungworm disease. Hawai’i Journal of Medicine & Public Health 72, 6–9.
Federspiel, F., Skovmand, S., Skarphedinsson, S., 2020. Eosinophilic meningitis due to Angiostrongylus cantonensis in Europe. International Journal of Infectious Diseases 93, 28–39. https://doi.org/10.1016/j.ijid.2020.01.012
Llaguno, G., Álvarez, S.F., Moran, W., Colcha, A.C., Castro, R.R., Reinoso, G.L., Ormeño, C.Á., Soria, G.M., 2020. Salud Pública Veterinaria: Desde la docencia hacia la investigación. Grupo LUZUMA.
